# The Direct Effects of Climate Change on Tench (*Tinca tinca*) Sperm Quality under a Real Heatwave Event Scenario

**DOI:** 10.3390/ani14050778

**Published:** 2024-03-01

**Authors:** Ignacio Fernández, Ana M. Larrán, Paulino de Paz, Marta F. Riesco

**Affiliations:** 1Spanish Institute of Oceanography (IEO-CSIC), Centro Oceanográfico de Vigo, Subida a Radio Faro nº 52, 36390 Vigo, Spain; 2Aquaculture Research Center, Agro-Technological Institute of Castilla y León (ITACyL), Ctra. Arévalo, Zamarramala, 40196 Segovia, Spain; ita-largaran@itacyl.es; 3Cell Biology Area, Department of Molecular Biology, Universidad de León, Campus de Vegazana, sn, 24071 León, Spain; ppazc@unileon.es

**Keywords:** thermal stress, aquaculture, germ cell, ROS, SOD, GPx, molecular assay

## Abstract

**Simple Summary:**

This study focuses on recreating more realistic models of global warming, particularly considering heatwave phenomena, in order to decipher its effects on fish male gametes (spermatozoa). Our results showed that currently occurring natural heatwave events decrease the progressive and total motility of sperm from tench males, increasing the ROS content and decreasing the gene expression of sperm quality markers. The present results confirmed the potential negative effect that a thermal stress model (by the occurrence of a heatwave) may have on fish spermatozoa quality and that it might compromise its fertilization capacity, not only hindering the sustainable farming of aquaculture species but also biodiversity conservation under current and realistic climate change scenarios.

**Abstract:**

Global aquaculture growth will most probably face specific conditions derived from climate change. In fact, the most severe impacts of these changes will be suffered by aquatic populations in restrictive circumstances, such as current aquaculture locations, which represent a perfect model to study global warming effects. Although the impact of temperature on fish reproduction has been characterized in many aspects, this study was focused on recreating more realistic models of global warming, particularly considering heatwave phenomena, in order to decipher its effects on male gametes (spermatozoa). For this purpose, thermal stress via a heatwave simulation (mimicking a natural occurring heatwave, from 24 to 30 °C) was induced in adult tench (*Tinca tinca*) males and compared with a control group (55.02 ± 16.44 g of average body wet weight). The impact of the thermal stress induced by this climate change event was assessed using cellular and molecular approaches. After the heatwave recreation, a multiparametric analysis of sperm quality, including some traditional parameters (such as sperm motility) and new ones (focus on redox balance and sperm quality biomarkers), was performed. Although sperm concentration and the volume produced were not affected, the results showed a significant deleterious effect on motility parameters (e.g., reduced progressive motility and total motility during the first minute post-activation). Furthermore, the sperm produced under the thermal stress induced by this heatwave simulation exhibited an increased ROS content in spermatic cells, confirming the negative effect that this thermal stress model (heatwave recreation) might have had on sperm quality. More importantly, the expression of some known sperm quality and fertilization markers was decreased in males exposed to thermal stress. This present study not only unveils the potential effects of climate change in contemporary and future fish farming populations (and their underlying mechanisms) but also provides insights on how to mitigate and/or avoid thermal stress due to heatwave events.

## 1. Introduction

Climate change is a reality, posing immediate and future threats to global food security. Global warming and native biodiversity loss are some consequences of climate change [1]. In this sense, among others, two main warming phenomena have been reported by experts: (1) a continued increase in global water temperatures, including ocean acidification, and (2) an increase in the frequencies and intensities of extreme heatwave events, defined as a time frame of very high temperatures over a sustained period of days. Continued emissions of greenhouse gases will lead to an increase in sea surface temperature (from 2 to 4 °C) throughout global marine waters by 2100. Moreover, a continuous increase in the duration, frequency, and intensity of extreme events (e.g., heatwaves) is also expected [1,2,3]. In general, marine and freshwater heatwaves will become stronger and more severe, especially towards the end of the century [4,5].

Aquatic populations in restrictive circumstances such as farmed fish in ponds, cages, and raceways or wild and stocked fish in lakes and reservoirs, streams, and rivers, will suffer the most severe effects of climate change. Aquaculture is considered a more feasible and sustainable production system for the continuous provision of animal protein to achieve food security and nutrition [6]. However, since most of the aquaculture production is performed in extensive, semi-intensive, and intensive systems (e.g., ponds, nets, and tanks) where the rearing temperature is not fully and tightly controlled and where the movement of animals to escape from suboptimal thermal regimes is restricted, the rapid expansion of global aquaculture will certainly be affected by the conditions derived from climate change [7], and aquaculture represents a perfect model to study global warming effects in aquatic environments [5]. Nevertheless, most of the current studies available in the literature have recreated specific thermal regimes that consider the projected temperature conditions of global warming but use contemporary models, neglecting the potential adaptation of these organisms to the gradual temperature increase [8,9,10]. Thus, for a more realistic and accurate prediction of the risk that climate change might directly impose on the farmed fish industry, controlled laboratory experiments that precisely mimic current temperature increase scenarios (e.g., heatwave occurrences) over contemporary specimens are required.

Temperature is an extremely important fish physiology driver, as many biological aspects can be affected by temperature in poikilothermic organisms (recently reviewed in [11]), including aerobic fitness, oxygen demand, hypoxia tolerance, growth rate, nutrition and feeding, immune response, disease proliferation, seasonal performance, pond stratification, maturation, and/or reproductive performance. All these factors can be decisive in the productivity and sustainability of fisheries and aquaculture [12]. Specifically, reproduction control is one of the key factors to warrant and increase aquaculture production through the domestication of new species, the intensification of already commonly produced species, and/or the breeding of particularly selected lines showing important biological and commercial traits (e.g., higher disease resistance, growth performance, stress tolerance, etc.) [13,14,15]. In this sense, adequate reproductive technologies aiming for control over the maturation process, sex differentiation, gametogenesis, and spawning are required. In all these processes, temperature acts as a regulatory factor on the hypothalamic–pituitary–gonadal (HPG) axis modulating sexual hormone synthesis. Nevertheless, although temperature increase is necessary to trigger many necessary reproductive processes (such as maturation, gametogenesis, ovulation, or spermiation), acute thermal stress at higher temperatures might provoke the opposite inhibitory effects on steroid hormones, resulting in gametogenesis failure [16].

Although knowledge on how the temperature effect in many aspects of fish reproduction (such as sex differentiation [17,18,19,20]) has been progressively gained, temperature impacts on their reproductive capacity have not been examined in detail, particularly in terms of gamete quality [21]. Lema et al. [22] recently reviewed the effects of temperature increase on fish reproductive biology, highlighting its effects on males at epigenetic, transcriptomic, enzymatic, and hormonal levels and their consequences: delayed sperm maturation, lower semen volumes, and reduced sperm motility. Since the final stages of gamete maturation are highly temperature-sensitive [10,23], the comprehension of global warming impact on gamete quantity and quality might not only allow for a deepening of our understanding on how climate change might impact fish farming but also for the identification, design, and implementation of effective strategies to avoid and/or decrease global warming impacts in aquaculture. Nowadays, despite previous skepticism on its paternal contribution to progeny [9,24,25], the importance of sperm cells and sperm-borne RNA content as vehicles for certain phenotype transmission to progeny is recognized [26]. Therefore, evaluating how thermal stress induced by naturally occurring heatwave events affects spermatozoa production and quality is needed.

Tench (*Tinca tinca*) is an extensively produced freshwater fish species in Europe, reared in extensive monocultures (ponds), and particularly appreciated in local regions of Spain and Italy as well as in European countries such as Poland, Germany, and/or the Czech Republic, being a highly demanded and valued gastronomic product [27]. Although widespread and considered a least concern species by the IUCN, tench is also currently facing decreases in population numbers in some parts of Europe [27]. Reasons for this population decline are the introduction of invasive fish species, degradation of spawning areas and habitats as a whole, eutrophication, and other anthropogenic factors such as global warming. Tench is a eurythermal species, a characteristic particularly relevant for deciphering how climate change might also affect fish species that show a wide tolerance to water temperature fluctuations.

This study aims to comparatively assess sperm production and quality when breeding males faced a heatwave—a more realistic and contemporaneous phenomena of global warming—with those maintained at an optimal and constant thermal regime during the breeding season. Specifically, we explored how a heatwave affects the sperm motility, REDOX balance, and gene expression of transcripts related to sperm quality and heat stress in order to unveil their consequences on spermatic cells. For this purpose, a particular thermal stress model mimicking a natural occurring heatwave has been reproduced, and potential consequences from a cellular and molecular prism in adult tench males were explored in order to confirm if heatwave events might impose a threat to the reproduction success of farmed fish.

## 2. Materials and Methods

### 2.1. Ethical Statement

The ARRIVE guidelines [28] were followed in all experiments. The Guidelines of the European Union Council (2010/63/EU) and Spanish regulations (RD 53/2013) were applied when handling animals. The Bioethical Committee from the University of León (OEBA-ULE-015-2022) and the Junta de Castilla y León approved all experimental procedures (Project License number: ULE015-2022). Technical work was carried out by persons with a FELASA class C permit for animal experimentation. The Center for Aquaculture Research is a research institution registered and authorized to perform animal experimentation (REGA number: ES401940000293).

### 2.2. Animal Housing

Fourteen male tench breeders (with 55.02 ± 16.44 g of average body wet weight and 12.54 ± 1.39 cm of total length) reared in captivity were randomly split into two experimental groups (*n* = 7 fish per condition) in ITACYL facilities (Castile and León, Spain), named the control (CTRL) group and heatwave (HW) group. Breeder fish were individually stocked into two 0.5 m^3^ indoor round fiberglass tanks, with a density of 0.72 and 0.81 kg/m^3^. The fish were maintained at 24 °C under a natural photoperiod. Water exchange (20% of total volume tank) was performed every day, and aeration was provided to keep dissolved oxygen levels above 6 mg/L and nitrogenous compounds (ammonium and nitrite water concentrations) below toxic values (0.01 and 0.05 ppm, respectively). A photoperiod simulation of the environmental conditions in the area was applied. Commercial pellets (Dibaq Aquaculture) were used to feed the broodstock (3% biomass per day). Levels of nitrogen compounds in water were monitored and maintained above toxic levels.

### 2.3. Heatwave Simulation 

Figure 1 shows a scheme of the experimental design performed. The experimental design was based on the maximum water temperature achieved (32.2 °C) in lagoons from the Iberian Peninsula [29], temperature increase in water bodies (6 °C [30]), and maximum heatwave length in the same region (14–25 days [31]).

Before thermal stress induction, the animals were maintained in experimental tanks for 15 days for acclimation, and sperm was extracted from all the animals in order to avoid spermatic residual cells from other spermatogenesis cycles. Adult tench males were exposed to a heatwave event (HW), characterized by a progressive daily increase in water temperature (1 °C per day) from 24 to 30 °C, with this high temperature maintained for 10 days, recreated from a single heatwave event previously registered in natural ponds in Tencas Mateo (a fish farm located in Segovia, Spain). At this point, 16 days after the start of the experiment, HW males were lightly anesthetized, and sperm was collected. For comparative purposes, adult tench males from the control group (CTRL) were not exposed to the heatwave event (HW), with their water temperature constantly maintained at 24 °C. Sperm was also collected from CTRL specimens at the same day the HW specimens were sampled, post start of the experiment.

### 2.4. Fish Growth Performance Monitoring and Sperm Collection

Fish body wet weight, total length, and condition factor analyses were performed at the end of the experiment (16 days after the start of the experiment). For this purpose, the fish were anesthetized with tricaine methanesulfonate (110 mg/mL). Length (in cm) was recorded as total length (TL), using a regular ichthyometer. Body wet weight was measured using a digital GRAM S3R-6KD balance (0.01 g). The condition factor (CF) was calculated as follows:CF = [weight (g)/TL3 (cm)] × 100

Subsequently, sperm was collected from the measured males. Tench sperm was collected from different experimental groups at 16 days after the start of the experiment, coinciding with the maximum peak of the heatwave, via abdominal stripping. Special care was taken in order to avoid sample contamination. In this sense, the genital area was previously cleaned and dried with absorbent paper. The samples contaminated with urine were discarded. Styrofoam support was used to store samples on ice until further analysis. The sperm was then diluted 10-fold in a non-activating medium: grayling immobilization solution (200 mm glucose, 40 mm KCl, 30 mm Tris, pH 8 [32]). Sperm quality was immediately analyzed using an SCA sperm analyzer and flow cytometry.

Spermatic samples were then centrifuged to seminal plasma collection (10 min, 800 g) and preserved at −80 °C until redox balance analyses were conducted. Phosphate buffered saline (PBS) was used to wash sperm pellets for 2 min, and a RNA later solution was used to conserve samples. Subsequently, they were stored at −80 °C for further molecular analysis.

### 2.5. Sperm Motility and Concentration Analyses

To determine sperm motility and kinetic parameters, sperm samples from different experimental groups were evaluated using the CASA system (computer-assisted sperm analysis) (Sperm Class Analyzer-SCA-6.3.0.59; Microptic S.L., Barcelona, Spain). A total of 1 μL of sperm was activated with 10 μL of freshwater supplemented with 1% BSA into a Makler counting chamber (10 μm depth; Sepi Medical Instruments, Mumbai, India) to assess sperm motility. A 10× negative phase contrast objective in a Nikon Eclipse Microscope (Nikon, Tokyo, Japan), equipped with a Baster A312fc digital camera (Baster Vision Technologies, Ahrensburg, Germany) was used. Motility parameters were considered at 15, 30, 45, and 60 s post-activation. These parameters were evaluated in order to capture at a rate of 100 frames/second, using pre-defined SCA fish sperm settings adapted for fish species. Four different fields with 400 sperm each were acquired. The absence of cellular contamination (with epithelial cells, leukocytes, and/or immature germ cells) was confirmed by evaluating their presence in the captured fields via SCA system. Total motility (TM), as the percentage of sperm with VCL (velocity according to the straight path) >15 μm/s (the threshold to consider sperm as motile); progressive motility (PM), as the percentage of sperm with VCL > 45 μm/s; and fast progressive motility (FPM), as the percentage of sperm with VCL > 75 μm/s, were registered.

### 2.6. Sperm Flow Cytometry Analysis

ROS content (mainly superoxide anion) was measured by employing the CellROX™ Deep Red probe. This allowed for the identification of spermatozoa with high superoxide anion content (sperm stained by CellROX™), as previously described by Riesco et al. [33]. Briefly, the sperm samples were diluted in PBS to obtain a total of 2 × 10^6^ spermatozoa per sample. The sperm samples were washed and centrifuged at 500× *g* for 10 min at room temperature. Then, cells were incubated at room temperature in the dark for 30 min with aliquots of CellROX™ (5 μM final concentration, Thermo Fisher, Madrid, Spain). To stop cell staining and avoid an overstaining effect, an additional wash was performed. Then, flow cytometry analysis was immediately conducted, with the pellet resuspended in 1 mL PBS.

Flow cytometry acquisition was performed in a flow cytometer (MACSQuant Analyser 10, Miltenyi Biotech, Madrid, Spain) controlled with MACS Quantify 2.13 software (Miltenyi Biotech, Madrid, Spain). Per sample, a total of 50,000 events and at least 20,000 sperm cells, at a flow rate of 200–300 cells per second, were acquired. FlowJo v.10.2 (Ashland, Wilmington, DE, USA) was used to analyze the data.

### 2.7. Spermatozoa RedoxSYS Analysis

Redox potential was measured using the RedoxSYS assessment (Luoxis Diagnostics, Inc., Englewood, CO, USA). The following two values were provided: (i) sORP reported in millivolts (mV), the integrated balance of oxidants and reductants in a specimen; and (ii) cORP expressed in microcoulombs (μC), the amount of antioxidant reserves. Sperm samples of the two experimental groups (CTRL and HW) were washed two times with PBS 1X. After that, 20 μL of the sperm samples (1 × 10^6^ sperm cells) were inserted into the RedoxSYS diagnostic system, and parameters were measured in triplicates. sORP (as mV/10^6^ sperm) and cORP (as μC/10^6^ sperm) average values were recorded and presented.

### 2.8. SOD and GPx Activities in Seminal Plasma

Glutathione peroxidase (GPx) and superoxide dismutase (SOD) activities were assessed in seminal plasma using Cayman kits (ref. 703102 and SD125, respectively). GPx activity was determined at 340 nm, and SOD activity was assayed at 505 nm [34]. Reactions were performed using technical triplicates. A BioTek Gen 5 microplate reader at room temperature was used to assess all enzymatic activities.

### 2.9. Gene Expression Analysis of Spermatozoa

TRIzol^®^ reagent (Fisher Scientific, Spain) was used according to the manufacturer’s indications to extract total RNA. Lysis was performed using plastic pestles for the mechanical disruption of tissues. The quality and purity of the samples were assessed with a NanoDrop™ One/OneC spectrophotometer (Thermo Fisher Scientific™, Spain). The isolated RNA showed high purity (A260/280 > 2.0), and no apparent degradation was observed in an agarose gel. Total RNA was subjected to a DNase treatment (RQ1 RNase-Free Promega, Madison, WI, USA) before reverse transcription (1 h at 37 °C) using the reverse transcriptase Moloney murine leukemia virus (M-MLV; Invitrogen, Madrid, Spain), with an oligo-d(T) primer and RNase OUT (Invitrogen, Spain), as specified by the manufacturer.

Additionally, 1 μg of the total RNA was reverse transcribed (RT) to complementary DNA (cDNA) with a High-Capacity cDNA Reverse Transcription Kit (Applied Biosystems, Madrid, Spain) following the kit guidelines. The cDNA samples and the negative control RT were kept at −20 °C until further use.

The reactions of a semi-quantitative PCR (qPCR) were performed on a StepOnePlus Real-Time PCR system (Applied Biosystems) in triplicate. The reactions consisted of 10 µL of SYBR Green PCR Master Mix (Applied Biosystems, Madrid, Spain), 1 µL of gene-specific primers (forward and reverse, 10 µM; Table 1), 6 µL of molecular grade water, and 2 µL of cDNA. A specific sample was run in each qPCR plate for calibration purposes [35]. Amplification reactions were as follows: 95 °C for 2 min, followed by 40 cycles at 95 °C for 10 s and 65 °C for 20 s. A melting curve was conducted for 15 s at 95 °C, 1 min at 70 °C, and 15 s at 0.5 °C increments until 95 °C were reached. Primers’ PCR efficiency was calculated via standard curve with 3 serial cDNA dilutions, while a non-template control and a negative control reverse transcription (RT) sample were also run in order to confirm that the amplification was not due to DNA contamination during RT and/or reaction mix preparation. The primers’ PCR efficiency for the genes finally used in qPCR analysis ranged 94.7–108.4%. Gene expression levels were determined according to Pfaffl’s mathematical method [36]. BestKeeper software version 1, “https://www.gene-quantification.de/bestkeeper.html (accessed on 23 October 2023)” was used to select the best reference gene for the study performed here. Based on the results of the geometric mean of expression levels in those genes, *glyceraldehyde-3-phosphate dehydrogenase* (*gapdh*) was selected as the best reference gene with the lowest coefficient of variation (4.86%) and standard deviation (1.24) compared to the other gene candidates (*rps11*, *tuba*, *rpl8,* and *ef1a*). A normalization procedure was then applied using this reference gene.

### 2.10. Statistical Analyses

Prism 8 (GraphPad Software, version 10, San Diego, CA, USA) was used for statistical analysis. Statistical methods were used to identify outliers, and all data were checked for normality (Kolmogorov–Smirnov test). With the data presenting a normal distribution (Gaussian distribution), a student *t*-test analysis was performed, while the Mann–Whitney test was used with non-parametric data. The results are presented as mean ± standard error of the mean (SEM). A total of 6–7 individual males were analyzed in each experimental group. In all the analyses, statistical significance was set at *p* < 0.05, and in the graphs, it will be shown as * *p* < 0.05, ** *p* < 0.01, and *** *p* < 0.001.

## 3. Results

### 3.1. Biometric Analyses of the Control and Heatwave-Exposed Male Fish

No mortality was recorded during the exposure to increasing temperatures that simulated the heatwave event. In addition, no significant differences in total length, body wet weight, and condition factors between both experimental groups after heatwave exposure were registered (Figure 2; Student’s *t*-test, *p* < 0.05). Mean values for total length, body wet weight, and condition factors ranged between 12.23 and 13.50 cm, 52.01 and 64.68 g, and 2.59 and 2.76, respectively.

### 3.2. Sperm Volume, Concentration, and Motility Parameters in Males Exposed to the Heatwave Event and the Control Group 

No alteration in sperm volume and concentration was detected in specimens subjected to a heatwave event nor was it detected in the control group (Figure 3A,B). However, tench males showed a significant (*p* < 0.05) deleterious effect concerning some motility parameters (Figure 3C–F) when exposed to a heatwave model mimicking natural and registered thermal conditions in Spanish freshwater basins. Progressive motility (PM) was reduced from 84.63 ± 7.16% in the control group to 55.72 ± 25.45% in the thermal stress group (HW) (Figure 2C; Student’s *t*-test, *p* < 0.05). Similarly, the total motility (TM) of sperm was lower in the heatwave-exposed males (71.21 ± 6.05 in HW versus 96.57 ± 2.02 in CTRL; Figure 3D; Student’s *t*-test, *p* < 0.05). By contrast, no significant differences were found in fast progressive motility when specimens from the control and heatwave groups were compared (Figure 3E). However, reduced sperm TM in HW specimens was maintained along the first minute post-activation in all the studied times (Figure 3F; Student’s *t*-test, *p* < 0.05) but not when PM was evaluated (Figure 3G).

### 3.3. Seminal Plasma Redox Balance and Spermatozoa Intracellular ROS Content in Males Exposed to the Heatwave Event and the Control Group

The thermal stress induced by the heatwave event led to a significant increase in the intracellular reactive oxidative species (ROS) content in spermatozoa cells, from 79.96 ± 3.47% in CTRL group to 91.13 ± 12.05% in HW group (Figure 4A; Student’s *t*-test, *p* < 0.05). Redox balance was also evaluated in seminal plasma by employing two approaches: through the quantification of enzyme activity related to redox balance, such as superoxide dismutase (SOD, Figure 4B) and glutathione peroxidase (GPX, Figure 4C); and a novel technology based on the measure of oxidation–reduction potential (Figure 4D,E). In both types of redox assays, no significant differences between both of the studied groups (CTRL and HW) were reported.

### 3.4. Gene Expression of Sperm Quality Markers in Males Exposed to the Heatwave Event and the Control Group

The gene expression of the *heat shock proteins* studied here (*hsp70* and *hsp90*) (Figure 5A,B) in the sperm of tench males exposed to the heatwave event was not significantly different from that of the control tench males. However, some sperm quality markers such as *bdnf* and *kita* were differentially expressed in these germinal cells when males were exposed to a heatwave event in comparison with the control group, with its expression being significantly downregulated in the HW group (Figure 5C,D).

## 4. Discussion

Although all organisms are able to adapt their metabolism to spatial and temporal changes in environmental temperature, such variations may have wide and deep consequences in different aspects of ectotherm biology [37], including behavior [38], feeding and digestive processes [39], skeletal muscle metabolism and mechanics [40], immunology [41], growth [42], and development [43,44], among others. Fish reproduction is not an exception. Among the reproductive processes already reported to be influenced by temperature (sex determination [17] and early puberty [10]), steroidogenesis, gametogenesis, and the final stages of gamete maturation seem to be highly temperature-sensitive (reviewed in [16]).

Physiological responses to thermal stress induced by climate change may be species-specific [24]. Until now, sperm quality variation after recreating global warming scenarios (increased average global temperature) has been evidenced in recent studies performed in salmonids [45,46], and a reduction in early-stage germ cell proportion (spermatogonia population) has been shown [47]. Thermally driven changes during spermatogenesis resulted in a lower number of spermatogonia, reducing the number of males undergoing spermiation [48]. By contrast, recent studies showed that temperature increases triggered spermatogonia proliferation and differentiation, as well as spermiogenesis, by increasing the abundance of spermatozoa in testes [49], and others indicated that semen volume is not impacted by temperature [47]. Therefore, effects on spermatozoa may depend on (i) the species considered and (ii) the duration and amplitude of temperature exposure (recently reviewed in [9]). Furthermore, it is crucial to accurately recreate more realistic and contemporaneous climate change scenarios in order to unveil the potential effects of climate change in fish reproductive biology, particularly in male fish.

Until now, the research studies that had been conducted only explored the effects of increased mean water temperature derived from the global warming process of climate change. In the majority of these studies, contemporaneous specimens (i.e., not exposed to temperature increases along different generations) were used, somehow neglecting their potential adaptation to this environmental condition. Due to climate change, the frequency and intensity of extreme events such as heatwaves is also predicted to increase [2]. These very high temperature events over a sustained period of days may affect contemporary specimens living in restrained conditions (e.g., fish farms, lagoons, etc.). The main objective of this work is to investigate how this short-term and more realistic climate change scenario affects the reproductive performance of male fish.

Tench (*Tinca tinca*) fish are an interesting model for exploring the effects of naturally occurring heatwaves, as they are currently reared in natural ponds (under restricted conditions that do not allow them to escape from changes in abiotic conditions such as high environmental temperatures and/or low dissolved oxygen). However, they also show several interesting physiological and biochemical features for studies, such as clear sexual dimorphism (in males, the pelvic rays are more robust and longer and extend beyond the anus) and tolerance to low dissolved oxygen values and to a wide range of environmental temperatures (up to 37 °C; [50]). Here, we demonstrated a deleterious effect on sperm quality from tench males exposed to a heatwave model mimicking the natural and registered thermal conditions during summer in inland Spanish lagoons, where they are reared [27,50]. The sperm motility of tench males exposed to a 16-day thermal stress model was reduced in terms of progressive and total motility along the first minute post-activation, confirming the negative effect previously reported in rainbow trout (*Oncorhynchus mykiss*), where sperm motility and concentration were reduced in male specimens who were exposed early on to a 90-day thermal stress model during the parr stage [51]. Considering the fact that rainbow trout is a temperate stenotherm [52] and that tench is an eurytherm species [50], the present results further highlight how currently occurring extreme events like heatwaves may result in reduced reproductive performance in male fish.

Bearing these findings in mind, we decided to elucidate the mechanisms triggering this disturbance in sperm motility in heatwave-exposed individuals. It is widely known that thermal stress provokes different levels of damage, including an increase in lipid peroxidation, potential mitochondrial membrane impairment, and high ROS production [53]. The spermatozoa are particularly susceptible to oxidative stress [54] due to their inadequate cell repair systems and/or insufficient antioxidant defenses. Indeed, high contents of ROS are capable of producing cell apoptosis, DNA strand breakages, mitochondria function impairment, and changes in membrane composition due to sugars, lipids, and amino acid oxidation, which might affect sperm fertilization ability later on. Although redox global status can be inferred by blood plasma analysis, it does not necessarily translate into a redox imbalance in sperm, as ROS generation can vary between subcellular compartments [55]. Moreover, blood plasma extraction would require the further handling (and perhaps harming) of animals. Thus, we analyzed the specific compartment related to reproductive performance (the sperm) here, which is more experimentally ethical, reducing the potential suffering of the animals. The imbalance between ROS levels and physiologic antioxidant levels can trigger oxidative stress responses with a consequent loss of sperm motility and decreased fertilization capacity [56,57]. Nevertheless, the antioxidant capacity of semen can vary among fish species [58], and the formation of ROS can cause different types of damage. In the first approach, we decided to apply flow cytometry techniques to detect ROS in the spermatozoa of the exposed and unexposed males. Our results confirmed that the ROS content increase in spermatic cells from the HW experimental group might be responsible for their impaired motility. Knowing the vital role of seminal plasma in protecting the spermatozoa against oxidative stress [59], we explored if redox balance in seminal plasma was also disrupted in the second approach.

No significantly reduced glutathione peroxidase (GPx) and superoxide dismutase (SOD) enzyme activities were observed in seminal plasma from HW males when compared to that of unexposed males. Although both GPx and SOD are master enzymes in vertebrates’ sperm redox balance to protect spermatozoa from oxidative damage [60,61], contradictory results have been reported in the literature on how redox enzymes respond to thermal stress, depending on the species considered, the biofluid, and the thermal stress induced. For instance, SOD plasma activity was not affected in two Antarctic notothenioid species subjected to a short thermal stress period [24]. SOD and GPx activities increased during a hot humid season compared to winter and spring seasons in the seminal plasma of Karan Fries bulls (*Bos taurus*) [62] but decreased in Ossimi rams (*Ovis aries*) in the summer season when compared to the winter season [63], as well as in guinea pigs when exposed to a heat stress condition of more than a 7 °C difference for 60 days [64]. Furthermore, oxidative stress seems to not be correlated with decreasing activities of GPx and SOD enzymes in humans [65]. A variety of compensatory processes have been also described to maintain redox status under physiological controlled conditions [66]. An innovative and more unspecific redox balance assessment, the static oxidation reduction potential (sORP), also did not show significant differences between sperm from exposed and unexposed males. The MiOXSYS System allows for the detection of oxidative stress by measuring the oxidation–reduction potential (ORP), directly evaluating the redox balance between ROS and antioxidants. Thus, although there was high ROS content (shown by the flow cytometry analysis) in the sperm from exposed males, it might be balanced by the presence of antioxidants as evidenced by the lack of differences in (sORP). This was further supported by the lack of differences in the activity of SOD and GPx in seminal plasma to potentially counteract the decreased pool of antioxidant compounds. Therefore, although apparently contradictory, these three approaches allowed us to have a clearly defined situation; sperm from males exposed to an HW event might have high ROS content, but content in antioxidants might be enough to not need the activity of SOD and GPx.

Oxidative stress (e.g., overabundance of ROS or a deficiency of antioxidants) is known as one of the main causes of male infertility [67]. Since no conclusive results on the potential effect of a heatwave event on tench sperm quality were obtained regarding oxidative stress, we decided to conduct a molecular analysis to decipher when the reproduced heatwave event might decrease the sperm quality and deepen the intracellular stress response. In the last years, the idea that certain sperm mRNAs could act as good molecular markers of sperm quality has been significantly reinforced [57,68]. Although spermatozoa are transcriptionally inactive cells [69], these transcripts are remnants from spermatogenesis and could be relevant during fertilization and the subsequent early embryonic development [70]. In fact, previous studies employing molecular approaches on spermatozoa have revealed different clues of the importance of the sperm molecular signature in fish reproductive dysfunctions and progeny development [71,72]. Surprisingly, some of these transcripts seemed to be conserved from teleosts to humans [73,74].

In one hand, heat shock proteins (HSP) have been identified as responsive proteins of acute heat stress [75,76]. Specifically, hsp70 and hsp90 are the most sensitive chaperones of stress and, depending on the magnitude and duration of thermal stress in fish, can be modulated [77,78,79]. An acute temperature response includes the activation of pathways such as proteolysis through the ubiquitin–proteasome and could increase the synthesis of heat shock proteins [80]. The present results showed a lack of change in *hsp70* and *hsp90* at the transcriptional level in sperm collected during the peak of the heatwave event. In different species of salmonids (stenotherm species), a consistent increase in *hsp70* and *90* expressions in gills were registered when the fish were exposed to a sharper temperature increase (2–2.5 °C day^−1^ versus the 1 °C day reproduced here) and maintained at high temperatures for 5–7 days [81]. Discrepancies with the present results might be due to the different types of species compared (stenotherms versus eurytherms) and the tissue (highly responsive at the transcriptional level to environmental changes (the gills) versus less responsive (the spermatozoa)). A comparison with other studies using eurythermal species suggests that the lack of differences (induction) of *hps* expression in our exposed animals might be due to the fact that the time of sampling did not immediately occur after the 6 days of increasing temperature (from 24 to 30 °C), when acute thermal stress and its subsequent HSP response might be expected. For instance, higher *hsp70* expression was detected in large (approx. 60 g of body weight) spotted seabass (*Lateolabrax maculatus*) at 12 h post-thermal stress induction, but their levels were found to not be different from unexposed specimens at 3 days (72 h) under thermal stress, similar with the gene expression profile of hsp90 that, at 72 h, already showed decreased expression compared to that of the peak registered at 24 h [82]. Furthermore, since a higher expression of *hsps* under temperature stress is also correlated with antioxidant enzyme activity [83], the absence of differences in SOD and GPx activities in seminal plasma is in line with the lack of increase in the gene expression of *hsp70* and *hsp90* in exposed tench males. Further studies might be needed in order to confirm if an altered gene expression of *hsps* might occur in more reactive tissues to heat stress (e.g., liver or gills) than in the sperm cells of tench.

On the other hand, regarding sperm quality markers, *brain-derived neurotrophic factor* (*bdnf*) transcript levels are related to protective effects on sperm function due to their important role against oxidative damage, modulating the action of some antioxidant enzymes [84]. BDNF seems to induce the phosphorylation of the cAMP response element-binding protein (CREB) and consequently reduces cytochrome C release into the cytosol [85,86]. In addition, *kit receptor a* (*kita*), has been characterized as crucial for sperm fertilization, considering that its expression appears to be particularly correlated with sperm DNA integrity [73]. When we analyzed these specific biomarkers, we detected severely reduced gene expression of both transcripts in the thermal stress group. A decrease in *bdnf* transcripts in human males was correlated with reduced sperm motility and low spermatozoa count and therefore associated with some types of male infertility [86]. In the case of *kita* in a previous study, an aberrant downregulation of this transcript has been observed in early stages of zebrafish embryos, resulting from fertilization with bad-quality sperm [71]. Therefore, the downregulation of both transcripts suggests that the quality of sperm from tench exposed to a heatwave event naturally occurring in freshwater bodies from Spain is compromised. Nevertheless, further functional analysis in non-model fish species, as tench is still desirable, is necessary in order to elucidate the biological significance of gene expression alterations in sperm.

Finally, knowledge gained of this fish species might be also indirectly transferred to other eurytherm fish species widely produced in similar environments such as gilthead seabream (*Sparus aurata*) in southern Spanish “esteros” and Mediterranean Sea cages, and/or in stenotherm species from freshwater basins like rainbow trout (*Oncorhynchus mykiss*). Understanding the mechanisms behind temperature-related distribution patterns of animal species may help to better explain contemporary biogeographic patterning but also elucidate how species biodiversity and abundance will be affected by global warming, as well as how well they perform or adapt to warmer ecosystems [87]. In non-restricted geographical environments, species might be able to advance the timing of spawning and/or shift their latitudinal distribution, such as in the case of marine fish species [88]. However, in species with restricted movement like tench, inhabiting natural lagoons and ponds, the quality of the sperm might be monitored through an integrative approach in order to decipher if heatwave events might pose a threat to this species and/or to predict if they are able to cope with climate change scenarios, as well as how this adaptation is performed at the molecular level and transmitted to future generations. Furthermore, this present research work anticipates silent but broader and deeper transgenerational effects of heatwaves, such as the recently reported mass mortality events registered in warm water bodies such as the Mediterranean Sea [89].

## 5. Conclusions

The present results confirmed the potential negative effect that a thermal stress model (via heatwave occurrence) may have on fish spermatozoa quality and that this might compromise its fertilization capacity, not only hindering the sustainable farming of aquaculture species but also biodiversity conservation under current and realistic climate change scenarios. Currently occurring natural heatwave events have been shown to decrease the progressive and total motility of sperm from tench males, increasing the ROS content and decreasing the gene expression of sperm quality markers (*bdnf* and *kita*). Nevertheless, further studies will be required to study thermal-stress-related mechanisms and sperm transcriptomics in a wider (omic) manner. Only with an integrative and deep characterization of how heatwave events affect fish physiology, and particularly reproduction, can new enlightening strategies be defined in order to mitigate and/or avoid biodiversity loss due to climate change.

## Figures and Tables

**Figure 1 animals-14-00778-f001:**
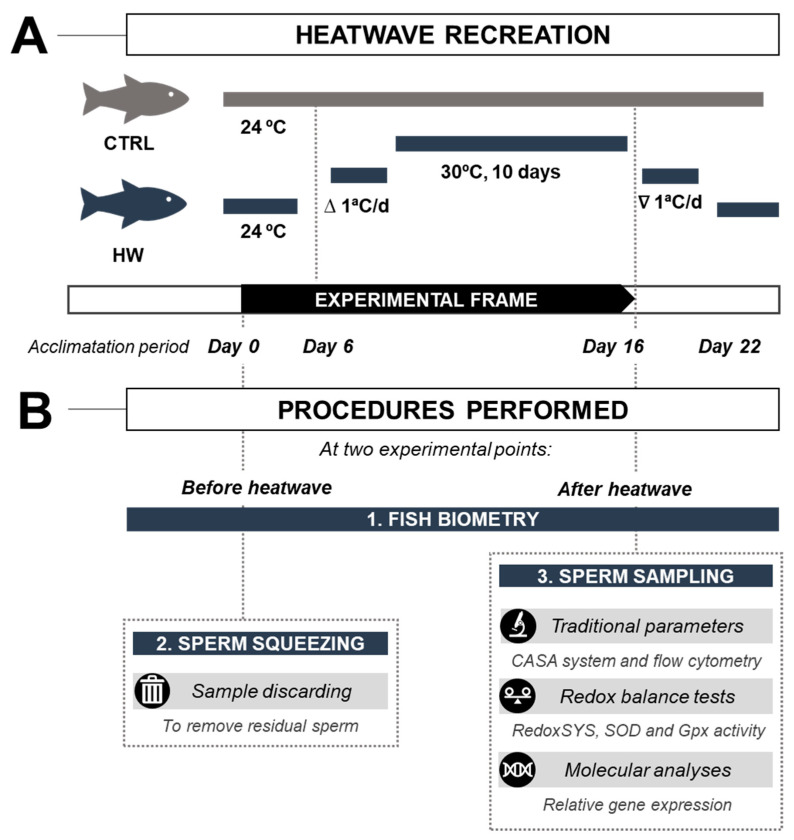
Experimental design performed in tench (*Tinca tinca*) males containing (**A**) a scheme of the heatwave recreation and performed analyses. (**B**) The thermal stress group (HW group) is represented in blue, and the control group (CTRL) is represented in grey. Seven males were analyzed in each experimental group.

**Figure 2 animals-14-00778-f002:**
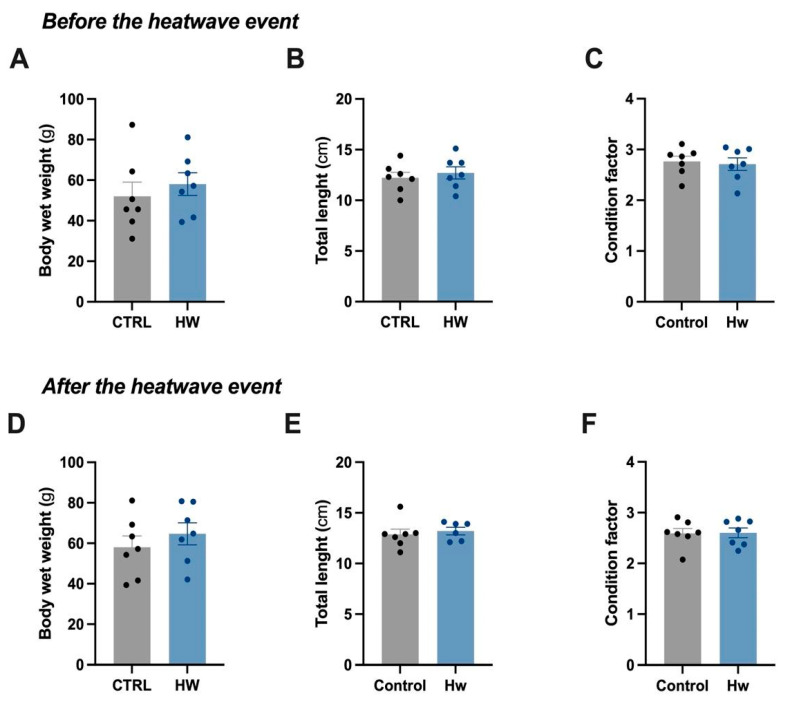
Biometric analysis of tench (*Tinca tinca*) males subjected to the heatwave event and the control group during spermatogenesis (detailed in the Methodology section). Dots and histograms in grey represent the control group (CTRL), not exposed to the heatwave event, while those in blue represent the thermal stress group (HW) exposed to the heatwave event. Six–seven males were assessed from each group. These biometric measures include body wet weight (**A**,**D**), total length (**B**,**E**), and condition factor (**C**,**F**) values ± standard error before and after the heatwave event.

**Figure 3 animals-14-00778-f003:**
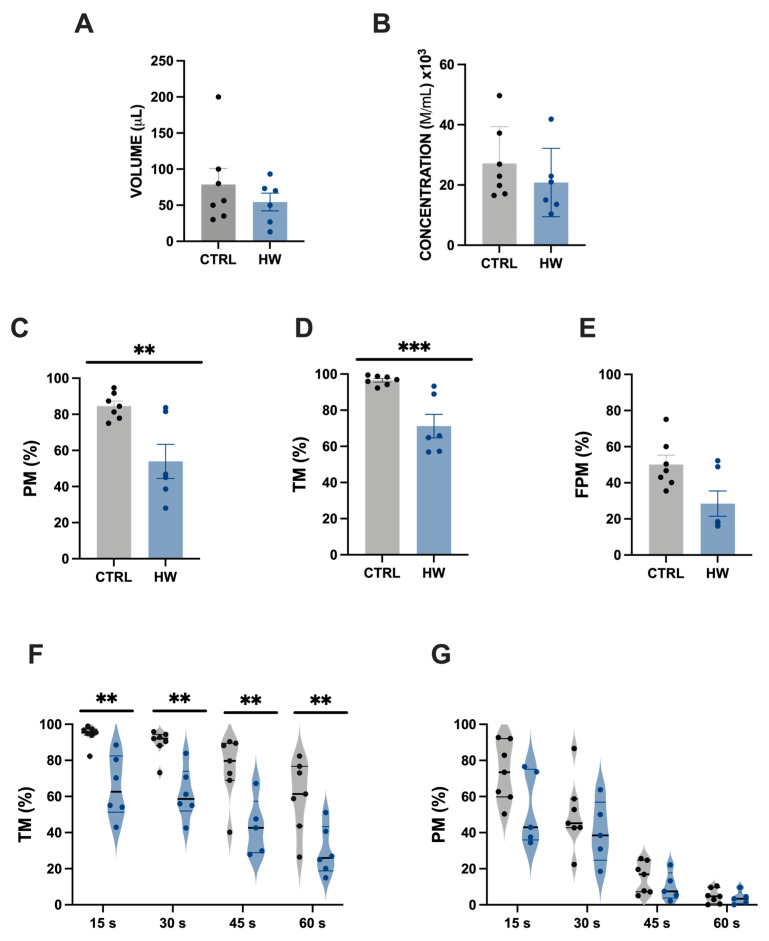
Traditional sperm quality analysis of tench (*Tinca tinca*) males subjected to the heatwave event and the control group during spermatogenesis (detailed in the Methodology section). Dots and histograms in grey represent the control group (CTRL), not exposed to the heatwave event, while those in blue represent the thermal stress group (HW) exposed to the heatwave event. (**A**) Sperm volume (uL); (**B**) spermatozoa concentration (M/mL); (**C**) progressive motility at the initial moment (PM, %); (**D**) total motility at the initial moment (TM, %); (**E**) fast progressive motility (FPM, %); (**F**) total motility at different post-activation times during the first minute; and (**G**) progressive motility at different post-activation times during the first minute. Six–seven males were assessed from each group. Graph dots represent individual males. Significant differences (*p* < 0.05) are represented with asterisks, ** *p* < 0.01 and *** *p* < 0.001.

**Figure 4 animals-14-00778-f004:**
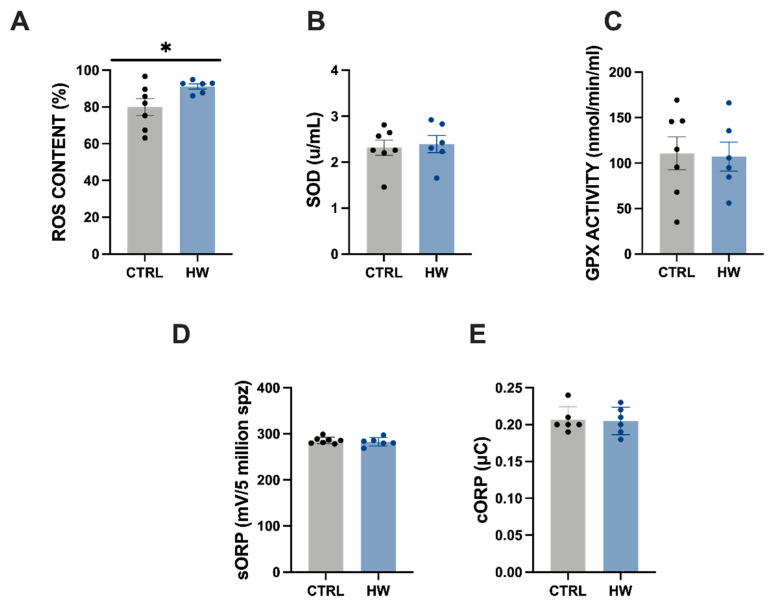
Redox analyses of tench (*Tinca tinca*) males subjected to the heatwave event and the control group during spermatogenesis (detailed in the Methodology section). Dots and histograms in grey represent the control group (CTRL), not exposed to the heatwave event, while those in blue represent the thermal stress group (HW) exposed to the heatwave event. (**A**) CellROX-positive cells (sperm ROS levels, %); (**B**) superoxide dismutase (SOD) activity (u/mL) in seminal plasma; and (**C**) glutathione peroxidase (GPx) activity (nmol/min/mL) in seminal plasma. (**D**,**E**) Static ORP (sORP) and capacitance ORP (cORP) indices (as mV/10^6^ sperm and μC/10^6^ sperm, respectively) as a measure of total redox balance. Six–seven males were evaluated in each group. Graph dots represent individual males. Significant differences (*p* < 0.05) are represented with asterisk.

**Figure 5 animals-14-00778-f005:**
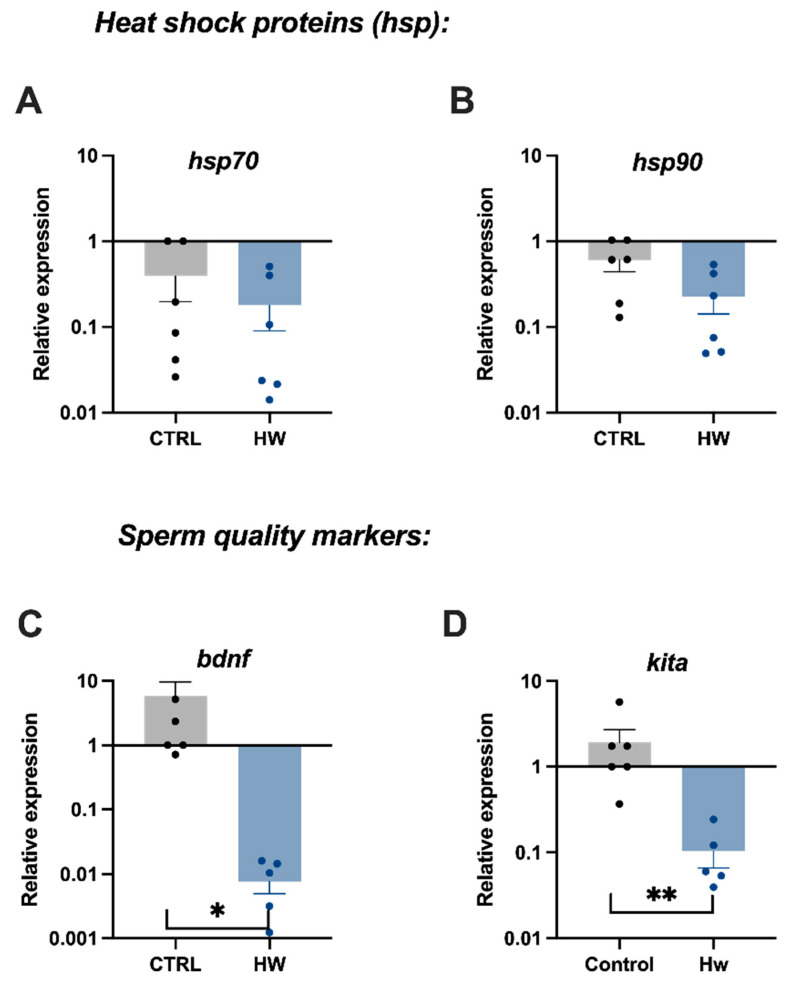
Relative gene expression in tench (*Tinca tinca*) sperm when males were subjected to the heatwave event and the control group during spermatogenesis (detailed in the Methodology section). Histograms in grey represent the control group (CTRL), not exposed to the heatwave event, while those in blue represent the thermal stress group (HW) exposed to the heatwave event. (**A**) The expression of *hsp70* and (**B**) *hsp90* genes. (**C**) The expression of *bdnf* and (**D**) *kita* genes. Six–seven males were assessed per group. Graph dots represent individual males. Significant differences (*p* < 0.05) are represented with asterisks, * *p* < 0.05 and ** *p* < 0.01.

**Table 1 animals-14-00778-t001:** Primers used for relative gene expression in tench (*Tinca tinca*) spermatozoa. Gene name, accession numbers (GenBank or Transcriptome Shotgun Assembly Sequence (TSA Seq)), primer sequences, and amplicon size are indicated.

Gene Name	Accession Number	5′ to 3′ Sequence	Amplicon Size (bp)
*Small ribosomal subunit protein uS17—rps11*	KX082695.1	F-CATCAGGCGGGACTACTTGC	86
		R- ATGGGGAGAGGTGGACAGACA	
*Tubulin alpha chain—tuba*	KX082693.1	F-GGCTGTGTTTGTAGACCTGGA	123
		R-CGGGCGTAGTTATTGGCT	
*Large ribosomal subunit protein uL2—rpl8*	KX082694.1	F-CTCCCACAACCCCGAGACCA	300
		R-AGACCGACCTTGCGTCCAGC	
*Elongation factor 1-alpha 1—ef1a*	KX082697.1	F-TGGAGACAGCAAGAACGACCC	186
		R-CGAGCACTGGAGCGTAGCCC	
*Glyceraldehyde-3-phosphate dehydrogenase—gapdh*	KX082698.1	F-CGTGTCCCCACCCCCAATGT	191
		R-GCAGCCTTGACCACTTTCTTGATGT	
*Heat shock protein HSP 90-alpha 1—hsp90a*	GFZX01157613.1; FZX01157620.1; GFZX01157619.1; FZX01157615.1	F-ACCCCCTCCATCCCATCGTC	264
		R-TGTCCTCATCGCCCTCCAGA	
*Heat shock 70 kDa protein—hsp70*	GFZX01034562.1; FZX01034564.1; FZX01034563.1; GFZX01034565.1; GFZX01252235.1; FZX01252238.1; GFZX01252236.1	F-ACGCAACACCACCATCCCCA	99
		R-GCCCTCTCTCCCTCATACACCTG	
*Mast/stem cell growth factor receptor kita—kita*	GFZX01104841.1; GFZX01104842.1; GFZX01056577.1; GFZX01056627.1	F-TTGGAAAGGTGGTTGAGGC	91
		R-TGGGCACTTGGTTTGAGCAT	
*Brain-derived neurotrophic factor—bdnf*	GFZX01217235.1	F-CACAGAGTGGGGGCGGGC	260
		R-GGCGGCGTCCAGGTAGTTTTT	

## Data Availability

Data will be available for all readers upon request.

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
