# Peer review of "The Direct Effects of Climate Change on Tench (Tinca tinca) Sperm Quality under a Real Heatwave Event Scenario"

_animals, 2024, doi:10.3390/ani14050778_

Round 1
Reviewer 1 Report
Comments and Suggestions for Authors
I am writing to provide my evaluation of the manuscript entitled "Direct effects of climate change on tench (Tinca tinca) sperm quality under a real scenario of a heatwave event" submitted for consideration in ANIMALS.
The study focuses on the effects of heatwave events on the sperm quality of tench fish. It found that natural heatwave events decrease the motility of sperm, increase the content of reactive oxygen species (ROS), and decrease the gene expression of sperm quality markers. This can potentially compromise the fertilization capacity of fish and hinder sustainable farming and biodiversity conservation under climate change scenarios. The study highlights the importance of accurately recreating realistic climate change scenarios to understand the potential effects on fish reproductive biology, particularly in males. It also discusses the role of oxidative stress and the potential use of sperm molecular markers to assess sperm quality. The study confirms an increase in ROS content in sperm cells exposed to a heatwave, which may be responsible for impaired motility.
Of course, there are some comments and suggestions for the manuscript that are given below:
-A review of previous studies on the effects of temperature stress on reproductive physiology, especially in males, is necessary.
-At the end of the introduction, the purpose of the study should be more detailed and clear.
-Isn't it better to use the term thermal stress instead of heatwave in the text? What do you mean by heatwave event?
- This paragraph “Adult tench males were exposed to a heatwave event (HW) characterized by a progressive daily increase of water temperature (1 °C per day) from 24 to 30 ºC, and maintaining at this high temperature during 10 days, recreating a single heatwave event previously registered in the natural ponds from Tencas Mateo, a fish farm located Segovia, Spain)” is ambiguous; Were the fish exposed to 30 ºC for 10 days?
- The scientific names of fish species should be in italics in all the text, for example in the captions of figures 3 and 5 and in the discussion.
- In Table 1, the primers of 9 genes are given, while only 4 genes are mentioned in the results. Why? And what was the basis for choosing these genes? It is better to talk about these gene indices in the introduction section.
-What role does oxidative stress play in sperm quality? For a better and more accurate analysis of antioxidant and ROS indicators, why was blood taken from fish not done? Sperm plasma cannot reflect the oxidative stress status of fish.
-In this study, the gene expression of heat shock proteins (hsp70 and hsp90) in the sperm of tench males exposed to a heatwave event was not significantly different from that of control tench males. Given that these genes are indicators of thermal stress; why didn't they change? Can't the reason be the late sampling of the fish (End of the tenth day)? And fish is somehow adapted to the stess. Couldn't the reason be that blood plasma or even tissues such as liver were not used for gene expression? Discuss these items in the discussion section.
Author Response
Reviewer #1:
- A review of previous studies on the effects of temperature stress on reproductive physiology, especially in males, is necessary.
Authors response: The literature on this issue is considerable, although most of them is considering the global warming approach and related to sex determination. Although to the best of our knowledge, the most recent and relevant studies on the effects of temperature stress on reproductive physiology (including males) has been cited along the present manuscript; we identified a very recent review on this issue. Lema et al. (2024) reviewed the effects of temperature increase on fish reproductive biology, and described the effects at epigenetic, transcriptomic, enzymatic and hormonal levels and their consequences: delayed sperm maturation, lower semen volumes, reduced sperm motility. For potential further interested readers in this issue the present reviewed is highly recommended. A single sentence has been added in the Introduction section: “Lema et al. (2024) recently reviewed the effects of temperature increase on fish reproductive biology, highlighting the effects on males at epigenetic, transcriptomic, enzymatic and hormonal levels and their consequences: delayed sperm maturation, lower semen volumes, reduced sperm motility.”.
Lema, S.C.; Luckenbach, J.A.; Yamamoto, Y.; Housh, M.J. Fish reproduction in a warming world: vulnerable points in hormone regulation from sex determination to spawning. Phil. Trans. R. Soc. B 2024, 379, 20220516. https://doi.org/10.1098/rstb.2022.0516
- At the end of the introduction, the purpose of the study should be more detailed and clearer.
Authors response: Authors agree with the reviewer that a more detailed description of the purpose of the study could be provided. In this sense, a sentence listing the analysis performed and the purpose of them has been now included: “Specifically, we explored how a heatwave affects sperm motility, REDOX balance and gene expression of transcripts related to sperm quality and heat stress in order to unveil their consequences on spermatic cells”.
- Isn't it better to use the term thermal stress instead of heatwave in the text? What do you mean by heatwave event?
Authors response: At the beginning of the Introduction the definition of a heatwave was already provided: “defined as a time frame of very high temperatures over a sustained period of days”. Reviewer can check Meehl and Tebaldi (2004) and IPCC (2023) to get a wider description of what are particular extreme events associated to the climate change, the heatwaves. Authors have discussed about the use of heatwave or thermal stress, and which one better describes the condition here experimentally recreated. Thermal stress is a very lax concept as it includes whatever conditions that induce a physiological response, no matter if it’s an increase or decrease in environmental temperature, how long it takes (acute or chronic: one single hour, 2 or more days, or months) and how progressive the change occurs (1 °C or 5 °C per hour or day). In the present study, authors consider that we are dealing with a particular example of thermal stress, the one related with a heatwave. That means, a progressive increase in water temperature and maintained during more than 3 days. Therefore, authors consider that heatwave should be the precise term used along the manuscript.
For further reading of heatwaves reviewer can read:
Meehl, G. A. y Tebaldi, C. (2004) “More intense, more frequent, and longer lasting heat waves in the 21st century”, Science, 305(5686), pp. 994-997.
IPCC. (2023) Intergovernmental panel on climate change. https://www.ipcc.ch/
- This paragraph “Adult tench males were exposed to a heatwave event (HW) characterized by a progressive daily increase of water temperature (1 °C per day) from 24 to 30 ºC, and maintaining at this high temperature during 10 days, recreating a single heatwave event previously registered in the natural ponds from Tencas Mateo, a fish farm located Segovia, Spain)” is ambiguous; Were the fish exposed to 30 ºC for 10 days?
Authors response: Yes, males of tench were exposed to 30 °C during 10 days, right after of a progressive increase of 1 °C per day in the water temperature (from 24 to 30 °C). Authors read the paragraph several times. Unfortunately, we were not able to identify where is the ambiguity in the description provided.
- The scientific names of fish species should be in italics in all the text, for example in the captions of figures 3 and 5 and in the discussion.
Authors response: Sorry for the mistake, how the scientific name of species was written has been correctedthroughout the manuscript.
- In Table 1, the primers of 9 genes are given, while only 4 genes are mentioned in the results. Why? And what was the basis for choosing these genes? It is better to talk about these gene indices in the introduction section.
Authors response: We apologize for the unclear information, 5 of the 9 genes are housekeeping genes (rps11, tuba, rpl8, ef1a and gapdh). Housekeeping genes (HKG; or reference genes) are needed to perform relative quantification. Please, see Pfaffl (2021) and Bustin et al. (2009) in this regard. Authors evaluated the suitability of 5 different genes as HKGs based on their stability and constitutive expression. Although their suitability has been already explored in tench (Panicz, 2016) for liver, spleen, gut, heart and muscle tissues in nutrigenomic studies, their suitability as HKG in sperm samples and under thermal stress was unknown. This is why the expression of these 5 genes was evaluated, to determine the most stable HKG using BestKeeper© Software. This concern has been clarified in the Material and Methods section.
Pfaffl, M.W. A new mathematical model for relative quantification in real-time RT–PCR. Nucleic Acids Res. 2001, 29, e45–e45.
Bustin, S., Benes, V., Garson, J.A., Hellemans, J., Huggett, J., Kubista, M., Mueller, R., Nolan, T., Pfaffl, M., Shipley, G., Vandesompele, J., Wittwer, C.T., 2009. The MIQE guidelines: minimum information for publication of quantitative real-time PCR experiments. Clin. Chem. 55, 611–622.
Remigiusz Panicz. 2016. Validation of reference genes for RT-qPCR analysis of growth hormone receptor and growth hormone expression in the tench (Tinca tinca) fed substituting poultry meal for fish meal. Aquaculture, 465, 179:188.
- What role does oxidative stress play in sperm quality? For a better and more accurate analysis of antioxidant and ROS indicators, why was blood taken from fish not done? Sperm plasma cannot reflect the oxidative stress status of fish.
Authors response: We totally agree with the Reviewer 1, sperm plasma cannot reflect the global oxidative stress status of fish. Nevertheless, this was not the aim of this study, since we were focused on the potential effect of heatwaves on sperm production and quality, regardless of the global fish status. The spermatozoa are particularly susceptible to oxidative stress (recently reviewed in Aitken and Drevet, 2020). This is mainly due to their inadequate cell repair systems, as well as insufficient antioxidant defenses due to very little cytoplasmic content. They are susceptible to lipid peroxidation (LPO) due to the high content of poly-unsaturated fatty acids (PUFAs) in their plasma membrane, resulting in the disruption of membrane permeability, and thus efflux of ATP, impairing flagellar movement. Indeed, high contents of ROS are capable of producing cell apoptosis, DNA strand breakages, mitochondria function impairment, and changes in membrane composition due to sugars, lipids, and amino acid oxidation, affecting at later times sperm fertilization ability.
However, the existence (or absence) of a redox imbalance in blood plasma does not necessarily translate into a redox imbalance in sperm (or safety). Redox status in plasma reflects a global disturbance of redox balance, but Reactive Oxygen Species (ROS) generation varied between subcellular compartment (Kaludercic et al., 2014). It might be consequence of an altered redox status in some organs (e.g., liver or gills), but not in all organs/tissues. Thus, we decide to specifically perform a redox status analysis on sperm, and avoiding the extra handling (and potential harming) of the animals in order to get blood plasma.
A small text has been added to the Discussion section to reflect the convenience of evaluating redox status in semen rather than in blood plasma. The text is as follows: “The spermatozoa are particularly susceptible to oxidative stress [51] due to their inadequate cell repair systems and/or insufficient antioxidant defenses. Indeed, high contents of ROS are capable of producing cell apoptosis, DNA strand breakages, mitochondria function impairment, and changes in membrane composition due to sugars, lipids, and amino acid oxidation, which might affect sperm fertilization ability later on. Although, redox global status can be inferred by blood plasma analysis, it does not necessarily translate into a redox imbalance in sperm, as ROS generation can varied between subcellular compartments [52]. Moreover, blood plasma extraction would require a further handling (and perhaps harming) of animals. Thus, we here analyzed the specific compartment related to reproductive performance (the sperm) and being more experimentally ethic, reducing the potential suffering of the animals”.
- In this study, the gene expression of heat shock proteins (hsp70 and hsp90) in the sperm of tench males exposed to a heatwave event was not significantly different from that of control tench males. Given that these genes are indicators of thermal stress; why didn't they change? Can't the reason be the late sampling of the fish (End of the tenth day)? And fish is somehow adapted to the stress. Couldn't the reason be that blood plasma or even tissues such as liver were not used for gene expression? Discuss these items in the discussion section.
Authors response: Authors have made the same question as reviewer #1, why the expression of HSP here evaluated didn’t change if those are indicators of thermal stress? Please, see our text in the discussion section regarding the sampling time and the potential adaptation of fish to the higher rearing temperature: “Comparison with other studies using eurythermal species, suggest that the lack of differences (induction) of hps expression in our exposed animals might be due to the fact that the time of sampling was not immediately after the 6 days of increasing temperature (from 24 to 30 ºC), when an acute thermal stress and its subsequent HSP response might be expected”.
Regarding if the expression analysis of these genes was conducted in other tissues, we cannot neglect that this altered gene expression of hps might happen in other more reactive tissues (e.g., liver and gills). However, as above mentioned, the use of these tissues required the sacrifice of fish. We selected the sperm as the target tissue to decipher if heatwaves affect reproductive performance in a more ethical manner (avoiding the sacrifice of fish). Nevertheless, a small sentence has been added to the Discussion section to highlight the limitation of our study, not considering the potential effects of heatwaves over other tissues: “Further studies might be needed in order to confirm if an altered gene expression of hsps might occur in more reactive tissues to heat stress (e.g., liver or gills) than in the sperm cells of tench”.

Reviewer 2 Report
Comments and Suggestions for Authors
The research topic investigating impacts of climate change on aquatic species is highly relevant. However, some methodological details require further clarification.
Did the thermal stress experiment accurately reflect naturally occurring heatwave conditions in terms of temperature increases and duration? What evidence supports the heatwave simulation model used?
How were water quality parameters like dissolved oxygen, pH, ammonia monitored during the experiment? Did water quality remain suitable for fish health in both groups?
What were the normal/baseline sperm motility and quality parameters for tench? How were these established without a control group at the start?
How many fish were sampled in total between the two experimental groups? Was this number sufficiently large for statistical analysis of results?
What evidence supports the flows cytometry and molecular assays used to characterize changes in redox status and gene expression? Have these been validated for tench sperm analysis?
Did fish size or any other common biometric variables differ significantly between groups at the start? How was randomization and allocation of fish to groups performed?
How were sperm samples collected and handled to ensure consistency between fish and minimize variability? What controls were included to rule out contamination?
What were the specific primer sequences and target genes analyzed? Were these genes previously shown to be relevant indicators of sperm quality/functional changes in fish?
How were raw CASA motility data preprocessed and analyzed? What thresholds/parameters were used to categorize sperm as motile vs. non-motile?
For flow cytometry analyses, how were controls and gates set to distinguish stained from unstained sperm cells?
For qPCR analyses, how was PCR efficiency validated? What controls ruled out contaminating DNA and ensured reliability of expression data?
Were any histological analyses of testes conducted to link functional changes to underlying tissue/cellular changes?
What further experiments could help validate results and establish causality between thermal stress and observed parameters?
Author Response
Reviewer #2:
- The research topic investigating impacts of climate change on aquatic species is highly relevant. However, some methodological details require further clarification.
Authors response: We thank Reviewer 2 for the accurate evaluation of our inputs and for giving us very valuable suggestions. Within the new version of the manuscript, we have made changes according to his queries and we have highlighted the modifications in the text through track changes system.
- Did the thermal stress experiment accurately reflect naturally occurring heatwave conditions in terms of temperature increases and duration? What evidence supports the heatwave simulation model used?
Authors response: Authors consider that the present experimental design, including a heatwave event of 16-days (6 days with an increase of water temperature of 1 °C/day until reaching 30 °C, followed by a period of 10 days at 30 °C) is one of the strengths of the present study. In our case, heatwave events with a 6 °C net gain through a progressive increase of water temperature of 1 °C/day, have been registered in rivers from the Iberian Peninsula (Ferreira-Rodriguez et al., 2018). Tench is reared under extensive systems (natural ponds/lagoons) with low control and monitoring of environmental variables such as the water temperature. This is why our experimental design is based on the maximum water temperature reached in lagoons from the same geographical region, up to 32.2 °C (Álvarez-Cobelas and Fernández-López, 2013), being found specimens at 37 °C in southern countries (Coad 2023). Regarding the duration, heatwaves of 25-34 days were registered in Spain in 2015 and 2020 (Infobae, 2023). Therefore, we consider that our experimental design is a quite faithful recapitulation of the current thermal stress that tench is subjected to, and a conservative replication of future scenarios taking into account the current predictions of increased frequency and intensity for heatwave events (IPCC, 2023). A small text has been added to the Materials and Methods section in order to further describe our experimental design and how realistic can be considered by potential readers: “Figure 1 shows a scheme of the experimental design performed. Experimental design was based on maximum water temperature achieved (32.2 °C) in lagoons from the Iberian Peninsula (Álvarez-Cobelas and Fernández-López, 2013), temperature increase in water bodies (6 °C; Ferreira-Rodriguez et al., 2018) and maximum heatwaves’ length at the same region (14-25 days; Infobae, 2023)”.
Álvarez-Cobelas, M. and Fernández-López, J. A primer on the limnology of some Castilian wetlands (central Spain). 2013, 239 p. LimnoIberia nº 1. Madrid. https://bibdigital.rjb.csic.es/idurl/1/16350
Coad, B.W. Freshwater fishes of Iran. Species Accounts-Cyprinidae-Tinca. www.purethrotle.com/briancode/species%20accounts/Tinca.htm
Infobae, https://www.infobae.com/espana/2023/08/09/la-ola-de-calor-mas-larga-de-la-historia-de-espana-26-dias-de-temperaturas-disparadas/
- How were water quality parameters like dissolved oxygen, pH, ammonia monitored during the experiment? Did water quality remain suitable for fish health in both groups?
Authors response: Description of water quality has been further described in the new version: “Water exchange (20% of total volume tank) was performed every day and aeration was provided to keep dissolved oxygen levels above 6 mg/L and nitrogenous compounds (ammonium and nitrite water concentrations) below toxic values (0.01 and 0.05 ppm, respectively)”.
- What were the normal/baseline sperm motility and quality parameters for tench? How were these established without a control group at the start?
Authors response: The normal/baseline sperm motility for tench has been previously described in literature for another authors:
Gil Anaya, M.C., Calle, F., Pérez, C.J., Martín-Hidalgo, D., Fallola, C., Bragado, M.J., García-Marín, L.J. and Oropesa, A.L. (2015), A new Bayesian network-based approach to the analysis of sperm motility: application in the study of tench (Tinca tinca) semen. Andrology 3: 956-966. https://doi.org/10.1111/andr.12071
The values obtained by these authors are very similar with present ones (96.57 ± 2.02 in CTRL) registering total motility values close to 93.2% in the 6- to 17 seconds post-activation. In addition, motility variation and decrease along post-activation time confirm our obtained results in control males (A significant variability in the percentages of motile spermatozoa was also observed between the males and increased with time. These values ranged from 89.7 to 99.7% in the first time period, from 77.3 to 94.4% in the second time period, and from 35.3 to 73.3% in the third time period).
- How many fish were sampled in total between the two experimental groups? Was this number sufficiently large for statistical analysis of results?
Authors response: We sampled a total of 14 tench breeders (n=7 fish per condition). We try to follow the Reduction principle, that refers to methods that minimize the number of animals used per experiment to obtain information with a given precision and robustness. Moreover, to avoid animal sacrifice, the less invasive samples have been chosen to perform this work (please, see responses to reviewer #1). In this sense, spermatozoa samples were isolated instead of testes. The number of animals sampled per condition is similar to other studies regarding the thermal effect in sperm motility:
Miriam Fenkes, John L. Fitzpatrick, Holly A. Shiels, Robert L. Nudds; Acclimation temperature changes spermatozoa flagella length relative to head size in brown trout. Biol Open 15 July 2019; 8 (7): bio039461. doi: https://doi.org/10.1242/bio.039461
- What evidence supports the flow cytometry and molecular assays used to characterize changes in redox status and gene expression? Have these been validated for tench sperm analysis?
Authors response: Recently, novel CellROX fluorescent probes in different colors have been introduced as ROS markers detecting hydroxyl radicals and superoxide anions in sperm cells (https://www.thermofisher.com/order/catalog/product/C10422). These have been previously used:
Rodrigues MB. An Efficient Technique to Detect Sperm Reactive Oxygen Species: The CellRox Deep Red® Fluorescent Probe. Biochem Physiol Open Access. 2015;04(02).
De Castro LS, De Assis PM, Siqueira AFP, Hamilton TRS, Mendes CM, Losano JDA, et al. Sperm oxidative stress is detrimental to embryo development: A dose-dependent study model and a new and more sensitive oxidative status evaluation. Oxid Med Cell Longev. 2016;2016.
Waltham M. The Molecular ProbesVR Handbook—A Guide to Fluorescent Probes and Labeling Technologies 11th ed.
In sperm cells, superoxide anion, together with hydrogen peroxide (H2O2), is one of the crucial products of REDOX reactions. The production of this anion has been previously studied in cyprinid fish and correlated with a decrease in motility, velocity and DNA integrity of carp spermatozoa: Sandoval-Vargas, L., Silva Jiménez, M., Risopatrón González, J., Villalobos, E.F., Cabrita, E. and Valdebenito Isler, I. (2021), Oxidative stress and use of antioxidants in fish semen cryopreservation. Rev. Aquacult., 13: 365-387. https://doi.org/10.1111/raq.12479.
Concerning to gene expression, two sets of transcripts were analyzed:
- related to heat stress, heat shock proteins (HSP), hsp90 and hsp70. Both of them have been validated as suitable temperature stress biomarkers in aquatic organisms, which is involved in thermo tolerance as well as regulating the immune system (https://www.mdpi.com/2076-3921/12/7/1444)
- ii) sperm quality markers: bdnf and kita, previously validated by our group in different fish species: Danio rerio, Solea senegalensis, Sparus aurata… and in humans. Although no specific validation has been done in tench, genomic inference suggest that gene function has been conserved along evolution in fish species.
- Did fish size or any other common biometric variables differ significantly between groups at the start? How was randomization and allocation of fish to groups performed?
Authors response: Data regarding the mean and standard deviation of fish weight and length at the start has been provided (55.02 ± 16.44 g and 12.54 ± 1.39 cm; respectively). No statistical differences were reported among both experimental groups, neither before and after the heatwave (please, see Figure 2). Fish with similar weight and size were selected from a major breeder population in order to avoid potential differences among experimental groups, and were randomly distributed in experimental tanks. Text description has been slightly modified to highlight that a random distribution of specimens has been done.
- How were sperm samples collected and handled to ensure consistency between fish and minimize variability? What controls were included to rule out contamination?
Authors response: Specimens were anesthetized as described in Materials and Methods section. Once the absence of reflexes was confirmed, the animals were gently immobilized with two hands, the surrounding area of the urogenital pore was dried and sperm collection was performed by abdominal massage according to routine in house protocols.
To minimize variability and to avoid masking the thermal effects by other processes in the same spermatogenesis cycle, sperm was obtained from all males and discarded at 0 days. With this procedure potential artifacts by residual spermatic cells from other spermatogenesis cycles were avoided.
In order to avoid contamination of sperm samples by feces, urine or freshwater, the genital area was cleaned and dried with absorbent paper. If the sperm samples were contaminated with urine or freshwater, the motility would have been immediately activated and the sperm samples would not show movement one hour after sperm collection.Secondly, the ejaculate contains cells other than spermatozoa. These include epithelial cells as well as leukocytes and immature germ cells. Although, these cells can be more precisely identified and quantified by detecting peroxidase activity or the antigen CD45 in mammalian species, they can be easily identified by examining the captured fields for concentration estimation in the SCA system, where sperm cells are quiescent. Their concentration can be estimated as for spermatozoa and it is possible to establish a ratio of these cells to the number of spermatozoa. We confirmed the absence of this type of cell contamination by evaluating the captured fields (3 fields per sample) by SCA system. This information has been partly added in the Material and Methods Section.
- What were the specific primer sequences and target genes analyzed? Were these genes previously shown to be relevant indicators of sperm quality/functional changes in fish?
Authors response: Specific primer sequences for the amplification of the target genes were designed based on deposited mRNA sequences at NCBI. Please, see the corresponding text description at Materials and Methods (Section 2.9. Gene expression analysis in spermatozoa) and Table 1. Among target genes, hsp70 and hsp90 were previously shown to be specific markers of stress response due to heat/cold stress (please, see the discussion section in this regard and in particular the references of Yan et al. (2017) and Jeyachandran et al. (2023)).
Yan, J.; Liang, X.; Zhang, Y.; Li, Y.; Cao, X.; Gao, J. Cloning of three heat shock protein genes (HSP70, HSP90α and HSP90β) and their expressions in response to thermal stress in loach (Misgurnus anguillicaudatus) fed with different levels of vitamin C. Fish Shellfish Immunol. 2017, 66, 103–111, doi:10.1016/J.FSI.2017.05.023.
Jeyachandran, S.; Chellapandian, H.; Park, K.; Kwak, I.S. A Review on the Involvement of Heat Shock Proteins (Extrinsic Chaperones) in Response to Stress Conditions in Aquatic Organisms. Antioxidants 2023, Vol. 12, Page 1444 2023, 12, 1444, doi:10.3390/ANTIOX12071444.
Regarding bdnf and kita, several articles conducted in fish species already shown evidences of BDNF functioning in reproduction (Cacialli et al., 2018) and behavior (Lucon-Xiccato, et al., 2022) as in mammals. Furthermore, gene expression level of both bdnfand kita were found to be correlated with sperm quality in fish species (Guerra et al., 2013; Riesco et al., 2017, 2019).
Cacialli, P.; D'Angelo, L.; de Girolamo, P.; Avallone, L.; Lucini, C.; Pellegrini, E.; Castaldo, L. Morpho-Functional Features of the Gonads of Danio rerio: the Role of Brain-Derived Neurotrophic Factor. The Anatomical Record 2018, 301, 140-147.
Lucon-Xiccato, T.; Montalbano, G.; Gatto, E.; Frigato, E.; D'Aniello, S.; Bertolucci, C. Individual differences and knockout in zebrafish reveal similar cognitive effects of BDNF between teleosts and mammals. Proc Biol Sci. 2022, 289:20222036.
Guerra, S.M.; Valcarce, D.G.; Cabrita, E.; Robles, V. Analysis of transcripts in gilthead seabream sperm and zebrafish testicular cells: mRNA profile as a predictor of gamete quality. Aquaculture 2013, 406–407, 28-33.
Riesco, M.F.; Oliveira, C.; Soares, F.; Gavaia, P.J.; Dinis, M.T.; Cabrita, E. Solea senegalensis sperm cryopreservation: New insights on sperm quality. PLoS One. 2017, 12:e0186542.
Riesco, M.F.; Valcarce, D.G.; Martínez-Vázquez, J.M.; Robles, V. Efect of low sperm quality on progeny: a study on zebrafish as model species. Scientific Reports 2019, 9:11192.
Discussion section has been changed accordingly to highlight the potential functional conservation of both genes in fish species, although further functional validation in non-model species is still desirable.
- How were raw CASA motility data preprocessed and analyzed? What thresholds/parameters were used to categorize sperm as motile vs. non-motile?
Authors response: At each time interval, three fields (including at least 400 spermatozoa each) were analyzed. Straight line velocity (VSL), curvilinear velocity (VCL), average path velocity (VAP), linearity (LIN = VSL/VCL), straightness (STR = VAP/VCL), beat-cross frequency (BCF), wobble (WOB), amplitude of lateral head displacement (ALH) and percentage of motile spermatozoa were the motility parameters assessed. The SCA settings were: 100 frames/s for acquisition. Total motility (TM), was assessed as the percentage of sperm with VCL (velocity according to the straight path) > 15 μm/s (the threshold to consider sperm as motile). The text in Materials and Methods section has been modified accordingly.
- For flow cytometry analyses, how were controls and gates set to distinguish stained from unstained sperm cells?
Authors response: Prior to performing the experiment, we have checked the emission spectra of CellROX™ Deep Red in order to confirm its fluorochrome emission spectra. In flow cytometry analysis, the population corresponding to the sperm was selected (gating), eliminating the rest of the events (debris) (confronting SSC-FSC). In addition, singlets were selected, eliminating doublets from the analysis (in a dot plot confronting SSC-height vs SSC area). Also, unstained control (blank tube) was used to determine positive and negative populations and to establish a threshold of staining. The protocol has been detailed described in the provided reference: Riesco, M.F.; Anel-Lopez, L.; Neila-Montero, M.; Palacin-Martinez, C.; Montes-Garrido, R.; Alvarez, M.; de Paz, P.; Anel, L. Proakap4 as novel molecular marker of sperm quality in ram: An integrative study in fresh, cooled and cryopreserved sperm. Biomolecules 2020, 10, 1–21, doi:10.3390/biom10071046.
- For qPCR analyses, how was PCR efficiency validated? What controls ruled out contaminating DNA and ensured reliability of expression data?
Authors response: Authors forget to include the description of how qPCR analysis was done, following the guidelines of minimum information for publication of Quantitative real-time Experiments (MIQE; Bustin et al., 2009). This include confirmation of PCR efficiency with the designed primers (ranging 94.7-108.4%), as well as controls to confirm that observed amplification was not due to DNA contamination of PCR. PCR efficiency has been calculated by standard curve with 3 serial cDNA dilutions, while a non-template control and a negative control reverse transcription (RT) sample were also run in order to confirm amplification was not due to DNA contamination during RT and/or reaction mix preparation. In the new version, authors include the missing information regarding primers PCR efficiency and DNA contamination controls in the Materials and Methods section. No amplification was obtained in wells with a non-template control or the negative control RT.
Bustin, S.A.; Benes, V.; Garson, J.A.; Hellemans, J.; Huggett, J.; Kubista, M.; … Wittwer, C.T. The MIQE guidelines: Minimum information for publication of quantitative real‐time PCR experiments. Clinical Chemistry 2009, 55, 611–622.
- Were any histological analyses of testes conducted to link functional changes to underlying tissue/cellular changes?
Authors response: As previously stated, the present study was designed in order to avoid the sacrifice of animals when exploring the effects of heatwave events on their reproductive capacity, as a first approach. Therefore, no tissue has been taken in order to conduct histological analyses. Nevertheless, present results opened a new research line in our lab, and ongoing work will further analyze the effects of heatwave events over fish reproductive capacity (including wider approaches such as RNA-Seq and histological analyses).
- What further experiments could help validate results and establish causality between thermal stress and observed parameters?
Authors response: As indicated in the previous comment, ongoing work has been designed in order to not only validate present results, but also to provide deeper knowledge. For instance, we are performing TUNEL analysis in histological sections of gonads in order to see if thermal stress led to increased cell apoptosis.
Taking into account the changes included in the present revised version according to the comments/suggestions from the two referees in order to improve the reading and the quality of the manuscript, authors hope its present revised version would be now suitable for publication in Animals.
Yours sincerely
Ignacio Fernandez
On behalf of the rest of co-authors

Round 2
Reviewer 1 Report
Comments and Suggestions for Authors
The all modification is fine.
Reviewer 2 Report
Comments and Suggestions for Authors
The authors have appropriately addressed the suggested revisions from the previous round of peer review. The article now meets standards for publication in this journal.